# Is the Association between Postpartum Depression and Early Maternal–Infant Relationships Contextually Determined by Avoidant Coping in the Mother?

**DOI:** 10.3390/ijerph18020562

**Published:** 2021-01-11

**Authors:** Cecilia Peñacoba Puente, Carlos Suso-Ribera, Sheila Blanco Rico, Dolores Marín, Jesús San Román Montero, Patricia Catalá

**Affiliations:** 1Department of Psychology, Rey Juan Carlos University, Avda. de Atenas s/n, 28922 Madrid, Spain; cecilia.penacoba@urjc.es (C.P.P.); sheila.blanco.rico@urjc.es (S.B.R.); jesus.sanroman@urjc.es (J.S.R.M.); 2Department of Basic and Clinical Psychology and Psychobiology, Jaume I University, 12071 Castellon, Spain; susor@uji.es; 3Hospital Universitario de Fuenlabrada, 28944 Madrid, Spain; dolores.marin@salud.madrid.org

**Keywords:** mother, avoidant coping, post-partum depression, pregnant women, early mother-infant relationship

## Abstract

This study analyzes the moderating role of avoidant coping (in early pregnancy) in the relationship between postpartum depressive (PPD) symptoms and maternal perceptions about mother–baby relations and self-confidence. Participants were 116 low-risk obstetric mothers (mean age = 31.2 years, SD = 3.95, range 23–42) who received care and gave birth at a Spanish public hospital. Measurements were made at two points in time: at first trimester of pregnancy (maternal avoidance coping) and four months after childbirth (PPD and maternal perceptions). Avoidant coping was associated with the perception of the baby as irritable and unstable (*p* = 0.003), including irritability during lactation (*p* = 0.041). Interaction effects of avoidant coping and postpartum depression were observed on the perception of the baby as irritable (*p* = 0.031) and with easy temperament (*p* = 0.002). Regarding the mother’s self-confidence, avoidant coping was related to a lack of security in caring for the baby (*p* < 0.001) and had a moderating effect between PPD and mother’s self-confidence (i.e., lack of security in caring for the baby, *p* =0.027; general security, *p* = 0.007). Interaction effects showed that the use of avoidant coping in the mother exacerbated the impact of PPD on the early mother–infant relationship.

## 1. Introduction

The process towards motherhood implies a vital change for women. This involves significant psychological, social, and physiological consequences, which ultimately increases their vulnerability for mental diseases [1]. For example, postpartum depression (PPD) prevalence rates can be established around 10–15% [2], even though estimates in low- and middle-income countries have been argued to be up to 25–20% of women [3].

PPD is associated with important consequences for both the mother and the newborn [4,5]. For the mother, it is manifested in symptoms such as low mood, reduced activity and energy, loss of enjoyment, reduced self-esteem, self-harm or suicide thoughts, sadness, guilt, or sleeping and eating disorders, among others [6,7,8]. In general terms, postpartum depression constitutes a serious public health problem associated with disability, comorbidity, and increased mortality through suicide [4].

In addition to the consequences of PPD on the mother, a topic of growing interest has been the evaluation of the consequences of PPD on the newborn. In particular, PPD is associated with maternal–infant interactions characterized by disengagement, hostility, and intrusion [9,10,11]. Mothers presenting PPD often exhibit negative thoughts about their confidence with baby care, such as deeming themselves to be insubstantial in taking care of the newborn [6,7,8]. Related to this, PPD can lead to poor infant feeding practices, which might result in baby malnutrition and reduced infant growth [12,13]. In addition, PPD can affect the mother’s capacity to bond with her new-born and depressed mothers may react negatively towards the child [14]. As a consequence of the previous, PPD has shown to negatively impact the cognitive and emotional development of the newborn later during infancy and childhood [15,16].

Given the relevance of PPD as a public health problem, the study of its risk variables is essential to establish preventive actions. As noted earlier, the relationship between PPD and negative maternal–infant interactions has been established. Nevertheless, this association is also complex and dynamic, and depends on the mother–infant interaction indicator that has been chosen [10]. For example, stronger effects have been found for PPD on behavioral child outcomes than on cognitive development such as language and intelligence quotient [10]. Furthermore, the effect of previous, chronic, or recurrent depression in the mother needs to be controlled, because in some reviews, it has been shown that they play a more relevant role than postpartum depression per se [10]. Finally, the relationship between maternal depression and early maternal–infant relationships might be moderated by maternal variables such as involvement in childcare; therefore, it would be necessary to adopt a multifactorial and prospective approach [5].

However, studies that investigate the impact of certain psychological variables in this relationship are limited [17]. Specifically, coping strategies constitute highly relevant variables within Lazarus and Folkman’s transactional model of stress and coping [18]. Coping is defined as a combination of “thoughts and behaviors used to manage the internal and external demands of situations that are appraised as stressful” [19]. Becoming a mother can certainly be a stressful situation, which requires the implementation of coping strategies. Numerous coping classifications exist. However, the distinction between approach and avoidance coping is probably the most widely used [20]. Approach and avoidance have been described as being “cognitive and emotional activity that is oriented either toward or away from threat”, respectively [21]. Approach coping involves reducing distress by taking steps to directly remove the stressor or reduce its impact, whereas avoidance coping intends to reduce distress by taking actions to avoid direct contact with the stressor [22]. Both strategies can be adaptive in the short term [23]. However, when exploring their influences in the long term, approach coping predicts improved psychological health and well-being, whereas avoidance has been related to a decrease in the outcome [24].

In this context, avoidance coping has stimulated greater interest in the literature owing to its maladaptive long-term effects on health-disease processes in both clinical and non-clinical populations [25,26,27]. Research has linked the use of avoidance coping with worse mental health, both during pregnancy and in the postpartum period [28]. Use of avoidance coping strategies is associated with decreased responsiveness to positive environmental stimuli [29]. Understanding coping energy and efforts as a finite resource, the investment of significant resources in avoidance may hinder efforts to engage in pleasurable activities, cultivate a mindful and present-focused demeanor, and recognize one’s innate strengths [30]. Avoidance of such thoughts and emotions may inhibit the individual from adopting more adaptive perceptions by preventing disconfirming evidence from being recognized or explored [31], thereby hampering the emergence of new learning experiences [24,32]. Studies have also investigated the mediating role of avoidant coping. For example, the use of avoidant coping was found to explain why perceived the neighborhood environment influenced depressive symptoms in pregnant African-American women [17]. What remains unclear is whether the implementation of avoidant strategies in combination with depressed mood can exacerbate the negative impact of PPD on the mother–baby relationship. This is important because avoidant coping is frequent, but not unique of depressed individuals. That is, not all mentally distressed individuals avoid to the same extent and individual differences in avoidant coping also exist in healthy individuals [33]. Therefore, the exploration of both their unique and combined impacts on baby–infant outcomes might provide new insights into therapeutic goals in this population (i.e., whether both constructs contribute independently to poor outcomes and whether their combinations lead to an even worse status).

Therefore, the present study focuses on the analysis of avoidance coping (in early pregnancy; first trimester) as a moderating variable in the relationship between PPD and early maternal–infant interactions (four months after delivery). The hypotheses are difficult to anticipate based on previous research. We expect, however, that PPD and avoidant coping will share variance in the prediction of baby–mother relationships. We also anticipate that the impact of PPD will be exacerbated by the use of avoidant coping when depressed.

## 2. Materials and Methods

### 2.1. Design and Sample

Women who received care and gave birth at the University Hospital of Fuenlabrada in Madrid, Spain, were recruited to participate in the study. The Ethics Committee of the Hospital approved the study protocol and the evaluation procedures. Women were eligible to participate if they met the following criteria: (a) were aged 18 years or older, (b) understood oral and written Spanish adequately, (c) did not have a multiple pregnancy, (d) did not have a serious or hormonal medical condition that could interfere with the pregnancy or dictate the type of delivery, (e) had a pregnancy that did not involve a severe fetal pathology, and (f) had not been diagnosed with a psychological disorder previously.

A prospective design was used. Women were assessed at two time points: first trimester of pregnancy and four months after delivery. At the beginning of pregnancy (during their first-trimester ultrasound), pregnant women that met the inclusion criteria were informed of the nature of the investigation and the conditions for their participation, and voluntarily agreed to participate. Women were excluded from the study if they completed the questionnaires improperly. At this first assessment (first trimester of pregnancy), a total of 287 pregnant women completed the avoidance coping measure. Of them, 116 women kept their participation four months after delivery. At this second assessment, both post-partum depression and maternal self-perception about early maternal–infant relationships measures were administered. Therefore, the final sample was composed of 116 low-risk obstetric mothers. The age of the participants ranged from 23 to 42 years (mean = 31.4; SD = 3.96). Half of them were new mothers (50.8%). Regarding educational level, 23.3% of the participants completed primary studies, 48.3% completed secondary studies, and 28.3% completed university studies. In total, 27.7% of the participants had history of at least one previous miscarriage. The majority of the women were employed during pregnancy (65%). Most pregnancies were planned (85%). The mean gestational age at delivery was 277.93 weeks (SD = 8.92; range, 253–299). The percentage of women who had vaginal, cesarean, and instrumental delivery was 62.5%, 17.9%, and 19.6%, respectively. Regarding the type of anesthesia, the women received epidural (66.4%), local (5.2%), spinal (7.8%), general (1.7%), and no anesthesia (18.9%). In spite of the high attrition rate, no significant differences were found for any of the sociodemographic or clinical variables considered between the initial sample (first trimester, *n* = 287) and the final sample (four months after delivery, *n* = 116): age (*p* = 0.38), educational level (*p* = 0.53), employed during pregnancy (*p* = 0.96), planning pregnancy (*p* = 0.49), previous pregnancies (*p* = 0.61), previous miscarriages (*p* = 0.87), gestational age at delivery (*p* = 0.78), type of delivery (*p* = 0.48), and type of anesthesia (*p* = 0.47). Regarding the outcome variables, no statistically significant differences were found between the samples for avoidance coping at the first trimester (*p* = 0.46).

### 2.2. Variables and Instruments

#### 2.2.1. First Trimester of Pregnancy

Avoidance coping. The avoidance the coping scale of the Coping Strategies Questionnaire (CAE) [34] was used. Items are rated on a five-point Likert-like scale ranging from 0 (never) to 4 (almost always). The CAE allows the assessment of seven basic independent styles of coping, including problem-solving coping, negative auto-focused coping, positive reappraisal, overt emotional expression, social support seeking, religious coping, and avoidance coping. The latter was used for the purpose of the present study. This scale contains seven items (e.g., “I tried not to think about the problem” or “I went to the movies, to dinner, for a walk, etc., to forget about the problem”). The minimum score for the avoidant subscale is 0 and the maximum is 24. Higher scores indicate higher avoidance coping. The questionnaire has been used in various studies and it has good indices of reliability and validity [35,36].

#### 2.2.2. Four Months after Childbirth

Post-partum depression. The Spanish version of the Edinburg Postnatal Depression Scale (EPDS) was used [37]. This scale consists of 10 multiple choice items (4 options each) concerning the respondent’s mood in the past 7 days. The total score can range from 0 to 30. Higher scores indicate higher PPD. Psychometric studies of the test have shown an alpha value of 0.87, a sensitivity of 85%, a specificity of 77%, and a positive predictive value of 83% [38,39].

Maternal perceptions about mother–baby relations. The “Mother and Baby Scales” scale (MABS) [40], which is a self-report instrument, was used to assess the mothers’ perceptions about her relationship with the newborn. Two types of maternal perceptions are particularly relevant in the MABS structure: (1) newborn behavior and (2) confidence in caring for the baby. The first (newborn behavior) contains 39 items that are grouped into five dimensions: Alert-Interest (8 items; “When I talk to my baby s/he seems to take notice”); Unstable-Irregular (15 items; “My baby has fussed before settling down”); Irritable During Lactation (8 items; “During feeds my baby has tended to fuss or cry”); Alert During Lactation (5 items; “After feeds my baby’s mood has been awake and alert”); and Easy Temperament (3 items; “Overall how difficult is your baby?”). Items are presented in a six-point Likert-type response format ranging from 0 (not at all/never) to 5 (a lot/frequently) for dimensions Alert-Interest, Unstable-Irregular, Irritable During Lactation, and Alert During Lactation, respectively, and 7 points for Easy Temperament. The second MABS dimension (i.e., maternal perceptions about confidence in caring for the baby) includes three subscales, namely, lack of confidence in care, lack of confidence in lactation, and general level of safety. These three areas are represented by 24 items: lack of confidence in care—13 items (“I’ve felt unsure whether I’ve been doing the right thing whilst looking after my baby”); lack of confidence in lactation—8 items (“I felt I haven’t always had enough milk to satisfy my baby”); and general level of safety—3 items (“Overall how stressful do you find it looking after your baby?”). A six-point Likert-type response format ranging from 0 (not at all/never) to 5 (a lot/frequently) is used in lack of confidence in care and lack of confidence in lactation. Seven points are used in the general level of safety scale. Studies so far support the validity of the MABS to measure maternal perceptions about mother–baby relations [40,41].

### 2.3. Procedure

This study is part of a larger investigation that aims to analyze the evolution of maternal psychosocial variables during pregnancy, delivery, and postpartum (FIS ISCIII PI07/0571).

A midwife participating in the study contacted the participants personally at the antenatal clinic during their first-trimester ultrasound. For eligibility reasons, she checked the electronic record to approach the women who met the inclusion criteria. These women then received information regarding the study and were invited to participate. In total, 320 women were found to be eligible during a period of 18 months. Of these, 287 women decided to participate (89.68%) and signed the informed consent form. After being enrolled in the study, the participants completed the avoidance coping instrument and returned it directly to the researcher. Four months after delivery, a questionnaire that included postpartum depression and maternal self-perception of both her baby and her own confidence as a mother was sent by mail. These were returned and correctly completed by 116 (39%) women from the initial sample.

### 2.4. Statistical Analysis

The SPSS 22 statistical package was used to perform all the analyses [42]. Descriptive analyses and internal consistency analyses (Cronbach’s alpha coefficients) were conducted as a first step. Means, standard deviations, and Pearson’s correlations were calculated for all study variables. Finally, the moderation analyses were carried out with model 1 of the PROCESS Macro version 3.4 [43]. PPD was set as the independent variable, maternal perceptions of early relationship and newborn behavior as dependent variables, and avoidant coping as the moderator. In the post-hoc analyses, non-centered variables were used to facilitate the interpretation of the results (centered variables were used elsewhere). Statistical significance was established at an alpha level of 0.05.

## 3. Results

### 3.1. Means, Standard Deviations, and Pearson Correlations between Study Variables

Table 1 shows the Cronbach alphas, means, standard deviations, and Pearson correlations between the study variables. The dimensions assessed show adequate values for internal consistency, and acceptable values in the case of the Alert-Interest—perception of the baby- (0.69) and lack of confidence in lactation (0.68). No significant associations were observed between avoidance coping at the end of the first trimester during pregnancy and PPD. Pregnancy avoidance coping was also unrelated to the mother’s perception of the baby or the lack of safety in lactation at the postpartum. Pregnancy avoidance coping, however, was significantly associated with less safety with baby care and general safety at the postpartum. Regarding the associations of PPD with postpartum maternal–infant relationships, a significant and negative association was observed between PPD and alert during lactation and with the general level of security. A positive association was found for PPD in relation to three mothers’ perceptions at postpartum (perception of the baby as Unstable-Irregular and Irritable During Lactation, and lack of self-confidence in lactation).

### 3.2. Moderation Analysis and Multivariate Linear Regression

The results of the regression analyses, including the moderation analyses, are presented in Table 2. Significant associations were observed between PPD and general safety level (Beta = −0.26, *p* < 0.001, 95% CI = [−0.37, −0.15]), lack of confidence in lactation (Beta = 0.39, *p* = 0.009, 95% CI = [0.10, 0.67]), lack of confidence in caring for the baby (Beta = 1.01, *p* < 0.001, 95% CI = [0.71, 1.32]), the perception of the baby as being alert during lactation (Beta = −0.19, *p* = 0.044, 95% CI = [−0.37, −0.01]), and the perception of the baby as being irritable during lactation (Beta = 0.32, *p* = 0.004, 95% CI = [0.10, 0.54]). Significant direct associations of avoidant coping were observed with the perception of the baby as being unstable (Beta = 0.95, *p* = 0.003, 95% CI = [0.33, 1.56]), the perception of the baby as being irritable during lactation (Beta = 0.31, *p* = 0.041, 95% CI = [0.01, 0.60]), and the lack of confidence in baby care (Beta = 0.81, *p* < 0.001, 95% CI = [0.37, 1.24]). Regarding the moderation analyses, the results revealed that avoidant coping moderated the relationship between PPD and the perception of the baby as being unstable (Beta = 0.16, *p* = 0.031, 95% CI = [0.01, 0.30]), easy temperament (Beta = −0.06, *p* = 0.002, 95% CI = [−0.09, −0.02]), lack of security in baby care (Beta = 0.11, *p* = 0.027, 95% CI = [0.01, 0.21]), and general safety level (Beta = −0.05, *p* = 0.007, 95% CI = [−0.09, −0.01]). The models accounted for 15%, 14%, 40%, and 28% of variance, respectively.

As noted earlier, post-hoc analyses were planned to analyze significant moderations in more detail. These were calculated and are presented in Table 3.

Figure 1 shows the moderation of avoidance coping in the relationship between PPD and the perception of baby as unstable. Specifically, the analyses indicated that PPD was positively associated with the perception of the baby as being unstable when avoidant coping was high (*p* = 0.011). When these were low, however, PPD and the perception of the baby as being unstable were no longer related (*p* > 0.05).

Figure 2 shows the moderation of avoidant coping in the relationship between PPD and the perception of the baby as having easy temperament. Specifically, the results indicated that PPD was negatively associated with the perception of the baby having an easy temperament when avoidant coping was high (*p* = 0.007). When avoidant coping was low, there was no relationship between PPD and mother perceptions about the baby (*p* > 0.05).

Figure 3 shows the moderation of avoidant coping in the relationship between PPD and lack of security in baby care. Specifically, the relationship between these variables increased with higher levels of avoidant coping (*p* < 0.001). The relationship between PPD and lack of safety in baby care, however, was significant across all avoidant coping levels.

Figure 4 shows the moderation of avoidant coping in the relationship between PPD and general safety level. Specifically, PPD was negatively associated with maternal safety when avoidant coping was high or medium (*p* < 0.001). When avoidant coping was low, the relationship between PPD and general safety was not significant.

## 4. Discussion

The aim of the present study was to explore the moderating role of avoidant coping during early pregnancy in the relationship between PPD and maternal–baby relations (i.e., perception of both the baby and herself and confidence as a mother). The relationship between postpartum depression and maternal–infant relationships has been investigated in the literature [5]. Consistent with past research [5,44,45,46], the results found in the present work revealed significant relationships between PPD and several dimensions of maternal–infant relationships (e.g., perception of the newborn as more unstable). As previous research shows, the association between PPD and maternal–infant relationships might not be linear [5]. In particular, the present investigation pointed to a coping strategy, namely avoidance, as a potential exacerbating factor for the relationship between PPD and maternal–infant relationships. These findings might be important for clinical practice as they point to a coping strategy in the mother that can be targeted to minimize the potentially harmful impact of PPD on maternal–infant relationships. Importantly, because avoidant coping was evaluated early during pregnancy, the findings might be important for prevention purposes. To the best of our knowledge, the exploration of the relationship between avoidant coping and early maternal–baby relationships, as well as its potential moderating role in the relationship between PPD and maternal–infant relationships, is new to the literature. Past research has previously pointed to the deleterious impact of avoidance coping, for example, for maternal stress levels during pregnancy and Apgar scores [47], satisfaction with childbirth [48], and maternal–fetal attachment [49]. This latter study showed that avoidant/disengagement coping and depression scores were negatively correlated with prenatal attachment scores [49]. In line with this study, our results showed that, even when the contribution of PPD was controlled for, avoidant coping in early pregnancy was significantly associated with several variables from maternal-infant relationships. Specifically, avoidant coping was related to the perception of the baby as unstable (in general and during lactation), and with the lack safety in baby care. These results are important as they point to a therapeutic target that might be included in interdisciplinary management programs during pregnancy when aiming to improve mother-infant relationships.

In addition to these direct associations between avoidant coping and mother–infant relationships, a key finding in the present study was that avoidant coping exacerbated the relationship between PPD and the mother–infant relationship. In particular, the moderation occurred with two factors associated with the perception of the baby and two factors related to maternal self-efficacy. In general terms, the results show that, when avoidance levels are high, the relationship between PPD and maternal–baby relationship increases and becomes statistically significant. A previous study [50] carried out among women with childhood trauma showed mediating effects of postpartum depression between trauma and maternal–infant bonding after controlling for covariates and antenatal distress. Maternal avoidance of fearful stimuli emerged as a potential affective mechanism. Although there are no similar previous studies in low obstetric risk samples to which our moderation results can be compared, our findings point in the same direction as previous research indicating the maladaptive role of avoidant coping in the gestation-puerperium process [47,48,49], including prenatal attachment [49].

A recent review indicated that the relationship between PPD and early maternal–infant relations is complex [5]. In this context, the contributions of certain variables to the aforementioned relationship have been studied, including gynaecological-obstetrics, demographic, infant, social support and parenting, and emotional availability factors. Among these factors, it is striking that the psychological factors related to the mother’s personality and coping strategies have received little attention [5]. The current investigation is a step in this direction. We hope that the findings will inspire researchers and clinicians alike.

In addition to the moderation analyses, the present study explored linear associations between PDD and maternal–infant relationships. A number of interesting findings were revealed. For example, a recent review indicated that lactation and maternal perceptions related to it are often omitted in research about the relationship between maternal depression and early baby–mother relations [5]. Our results, therefore, provide relatively novel data in this regard. Particularly, we evidenced that PPD is associated with the perception of the baby as being both less alert and more irritable during lactation. The relationship between breastfeeding problems and depression had been previously reported, especially in the first postpartum days [51,52]. However, the inclusion of maternal perceptions on the baby’s attitude towards lactation is new to the literature [53]. Even though the following is only hypothetical at this stage, it is possible that the relationship between PPD and breastfeeding cessation is mediated by perceptions in the mother that have been revealed to be associated with PPD in the present study (perception of the baby as being less alert and more irritable during lactation). These findings open new avenues for research and clinical practice.

Another finding in relation to PPD and maternal–infant relationships refers to the mother’s perception of self-efficacy. In the present study, a significant association emerged between depressive symptoms and the lack of security in baby care, the lack of security in lactation, and the general level of security. Self-efficacy, which refers to the confidence individuals have in their abilities to successfully perform their duties, such as parenting [54], has been found to be related to maternal depression and maternal behavior [55,56,57]. Not surprisingly, several studies have emphasized the role of early diagnosis and treatment of depressive symptoms and promotion of maternal self-efficacy to improve the overall functional status of mothers in the postpartum [57,58]. Maternal depressive symptomatology and self-efficacy have been directly and significantly correlated with infant self-regulation and maternal self-efficacy has been found to mediate the relationship between maternal depressive symptomatology and infant self-regulation [59]. High maternal efficacy has also been shown to buffer the detrimental impact of attachment insecurity and low self-esteem on maternal depression [60]. For example, previous research suggests that breastfeeding self-efficacy at the early postpartum period can predict postpartum depression [61]. In sum, both the present and past research suggest that maternal self-efficacy may be an important protective factor against depression [62].

An interesting finding was that avoidant coping and PPD were not related. The relationship between avoidance and depressive symptoms has been extensively analyzed in both clinical [63,64] and non-clinical samples [26]. Its study in the context of PPD has been less frequent, but studies appear to indicate modest, yet significant associations between avoidant coping in the mothers and PPD [65,66]. One possibility for the non-significant association between avoidant coping and PPD in the present study might lie in the different times at which both variables were collected (first trimester of pregnancy for avoidant coping and four months after delivery for depression). Another possible explanation for the discrepancy between the results could lay in the fact that previous studies have been carried out in high obstetric risk samples (mothers of infants treated in a neonatal intensive care unit) [65,66].

The present study has some limitations that should be mentioned. Regarding the generalizability of the results, for example, it should be noted that participants included a convenience sample made up of pregnant volunteers recruited through a health center. This may indeed limit its representativeness and attempt against the generalizability of these findings to the general population of pregnant women. On the other hand, it must be taken into account that our sample included women of low obstetric risk, which is likely to have resulted in relatively favorable PPD scores. Taking both shortcomings together, it would be interesting to explore to what extent the current study findings are replicated with samples presenting a higher PPD profile. Given the relationship between avoidant coping and PPD [28], it is possible that a reduced variability of avoidant coping scores in high PPD samples would result in a reduction moderating role of avoidant coping in the relationship between PPD and mother–infant relations. Further research is needed to shed light in this regard.

Despite the above limitations, as noted, this study might have some important practical implications. The fact that avoidance coping was evaluated early during the first trimester of pregnancy supports the idea that we could establish prevention strategies to improve the mother’s mental health and her relationship with the newborn as early as during the first trimester during pregnancy. Both emotional processing (via emotion-focused techniques integrated within cognitive-behavioral therapy) and cognitive restructuring (via overall cognitive-behavioral therapy techniques) have been shown to be essential elements in the therapy for depression. Assimilating emotion-focused techniques into cognitive-behavioral therapy (CBT) facilitates greater emotional processing during treatment and greater self-efficacy during follow-up. Additionally, cognitive restructuring in either standard CBT or exposure-based CBT leads to increased self-efficacy and reduced avoidant coping [67]. Important for the present study, the reduction of avoidance has been described as a key change mechanism in CBT for depression [68]. Therefore, according to the present and past research, the identification and intervention of avoidant coping early during pregnancy would be important for the promotion of the mother’s mental health in the postpartum period, as well as for her relationship with the newborn.

## 5. Conclusions

PPD is especially associated with the self-perception of security of the mother both in caring for her child and in lactation, but also in relation to the negative perception of the child during lactation. Maternal avoidant coping in early pregnancy has been associated with a lack of confidence in caring for the baby and the perception of the child as unstable and irritable during lactation. The moderating role of maternal avoidant coping in the relationship between PPD and the perception of the baby (unstable and easy temperament), along with the self-perception of security (in baby care and general safety level), are of special interest. When maternal avoidant coping was high, PPD was positively associated with the perception of the baby as being unstable and without an easy temperament. The relation between PPD and lack of safety in baby care increased with higher levels of avoidant coping, and PPD was negatively associated with maternal safety when avoidant coping was high or medium. Therefore, maternal avoidance coping in early pregnancy is suggested to be a risk factor moderating the complex and dynamic associations between PPD and mother–child relationships, in this case, through maternal perceptions both of the newborn and of their self-confidence as mothers.

## Figures and Tables

**Figure 1 ijerph-18-00562-f001:**
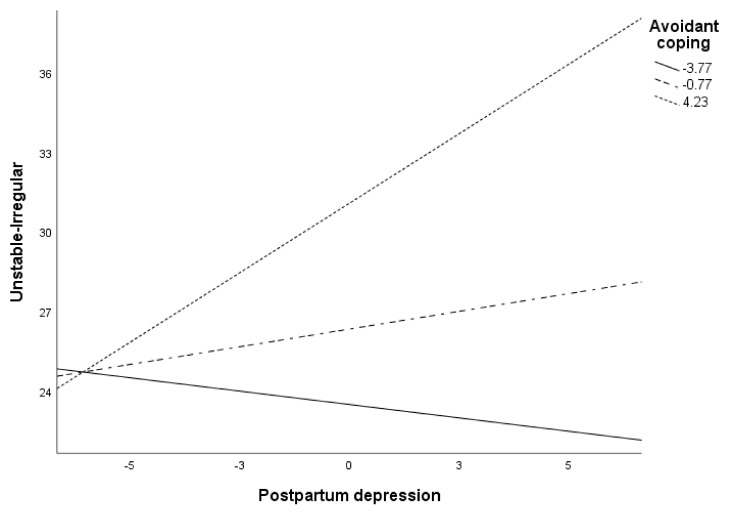
Conditional effects of post-partum depression (PPD) on perception of baby as unstable at values of avoidance.

**Figure 2 ijerph-18-00562-f002:**
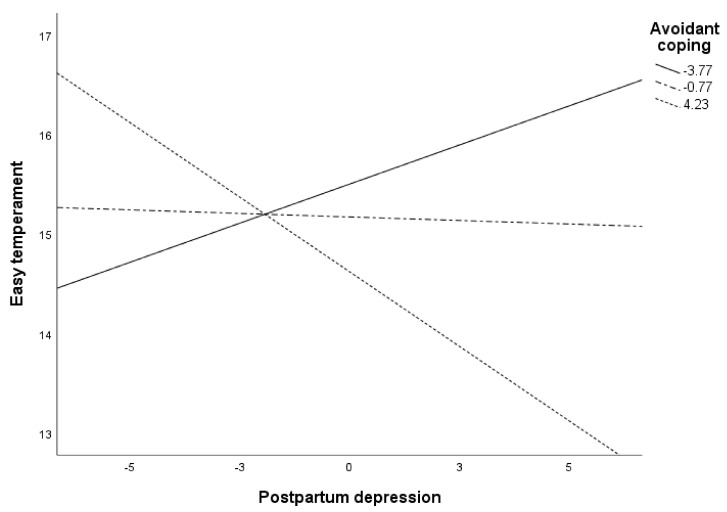
Conditional effects of PPD on perception of baby with easy temperament at values of avoidance.

**Figure 3 ijerph-18-00562-f003:**
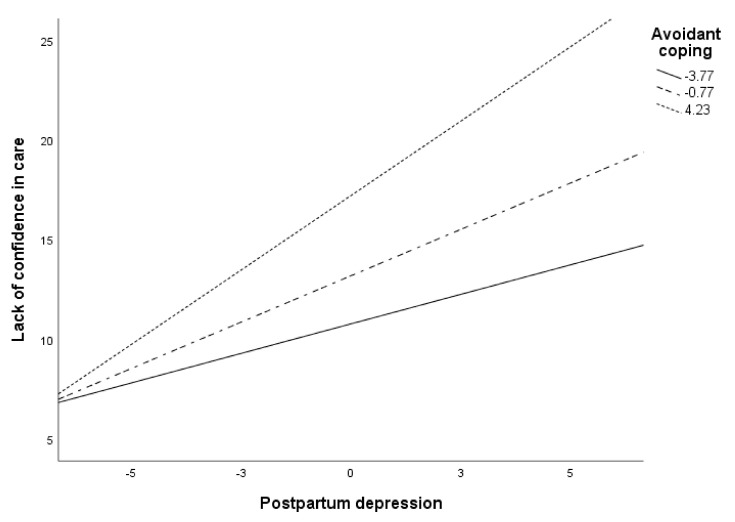
Conditional effects of PPD on lack of security in baby care at values of avoidance.

**Figure 4 ijerph-18-00562-f004:**
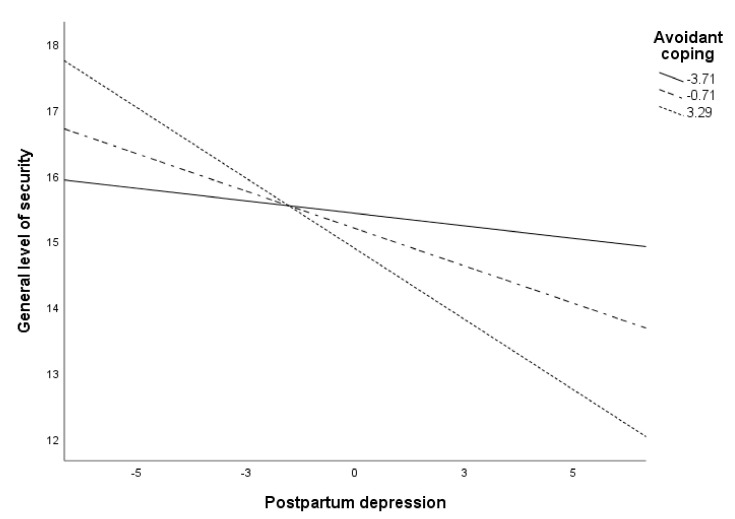
Conditional effects of PPD on maternal general level of safety at values of avoidance.

**Table 1 ijerph-18-00562-t001:** Means, standard deviations, and Pearson correlations between study variables.

Variable	Cronbach alpha	Mean (SD)	2.	3.	4.	5.	6.	7.	8.	9.	10.
1.Avoidance	0.72	9.15 (3.71)	0.09	0.06	0.13	−0.04	−0.16	0.13	0.30 **	0.08	−0.23 *
2. PPD	0.88	6.08 (4.86)		−0.13	0.22 *	−0.10	−0.25 **	0.30 **	0.51 **	0.30 **	−0.43 **
3. Alert-Interest	0.69	32.18 (4.64)			0.09	0.08	0.06	0.02	−0.12	−0.03	0.12
4. Unstable-Irregular	0.78	27.84 (11.92)				−0.74 **	-0.13	0.69 **	0.55 **	0.09	−0.46 **
5. Easy Temperament	0.80	14.85 (2.97)					0.07	−0.52 **	−0.39 **	−0.02	0.48 **
6. ADL	0.79	16.01 (4.63)						−0.05	−0.27 **	0.08	0.40 **
7. IDL	0.80	7.02 (5.38)							0.46 **	0.29 **	−0.37 **
8. LCC	0.78	14.73 (9.66)								0.25 **	−0.67 **
9. LCL	0.68	7.43 (6.22)									−0.21 *
10.GLS	0.81	15.03 (3.05)									

PPD: post-partum depression; ADL: Alert During Lactation; IDL: Irritable During Lactation; LCC: lack of confidence in care; LCL: lack of confidence in lactation; GLS: general level of safety. ** p* < 0.05; ** *p* < 0.01.

**Table 2 ijerph-18-00562-t002:** Prospective prediction of maternal perceptions about mother–baby relationships from postpartum depression, avoidance coping, and their interaction.

Variable	*R^2^*	*F*	*p*	Beta	*t*	*p*	95% CI
DV = A	0.08	2.29	0.083				
PPD				−0.17	−1.80	0.074	−0.36, 0.02
Avoidance				0.26	1.80	0.063	−0.01, 0.511
Interaction				−0.01	−0.40	0.689	−0.07, 0.05
DV = UI	0.15	4.84	0.003				
PPD				0.39	1.73	0.086	−0.06, 0.83
Avoidance				0.95	3.03	0.003	0.33, 1.56
Interaction				0.16	2.19	0.031	0.01, 0.30
DV = ET	0.14	3.72	0.015				
PPD				−0.06	−0.92	0.358	−0.18, 0.07
Avoidance				−0.11	−1.31	0.191	−0.28, 0.06
Interaction				−0.06	−3.10	0.002	−0.09, −0.02
DV = ADL	0.10	3.17	0.028				
PPD				−0.19	−2.04	0.044	−0.37, −0.01
Avoidance				−0.18	−1.40	0.164	−0.44, 0.08
Interaction				0.04	1.23	0.221	−0.02, 0.09
DV = IDL	0.16	5.14	0.002				
PPD				0.32	2.91	0.004	0.10, 0.54
Avoidance				0.31	2.06	0.041	0.01, 0.60
Interaction				0.07	1.90	0.060	−0.01, 0.14
DV = LCL	0.08	2.40	0.073				
PPD				0.39	2.66	0.009	0.10, 0.67
Avoidance				−0.06	−0.31	0.760	−0.46, 0.33
Interaction				−0.01	.20	0.845	−0.11, 0.09
DV = LCC	0.40	18.71	<0.001				
PPD				1.01	6.48	<0.001	0.71, 1.32
Avoidance				0.81	3.66	<0.001	0.37, 1.24
Interaction				0.11	2.25	0.027	0.01, 0.21
DV = GLS	0.28	8.67	<0.001				
PPD				−0.26	−4.74	<0.001	−0.37, −0.15
Avoidance				−0.08	−0.93	0.355	−0.24, 0.09
Interaction				−0.05	−2.73	0.007	−0.09, −0.01

PPD: post-partum depression; A: Alert-Interest; UI: Unstable-Irregular; ET: Easy Temperament ADL: Alert During Lactation; IDL: Irritable During Lactation; LCC: lack of confidence in care, LCL: lack of confidence in lactation; GLS: general level of safety; DV: Dependent Variable; CI: Confidence interval.

**Table 3 ijerph-18-00562-t003:** Conditional effects of PPD on maternal perceptions about newborn behavior and confidence in caring for the baby.

Variable	Avoidance	Beta (PPD)	*t*	*p*	95% CI
UI					
	−3.77	−0.20	−0.63	0.528	−0.83, 0.43
	−0.77	0.27	1.21	0.229	−0.17, 0.70
	4.22	1.05	2.59	0.011	0.24, 1.85
ET					
	−3.77	0.16	1.86	0.067	−0.01, 0.33
	−0.77	−0.01	−0.23	0.819	−0.14, 0.11
	4.22	−0.30	−2.75	0.007	−0.52, −0.08
LCC					
	−3.77	0.59	2.64	0.009	0.15, 1.03
	−0.77	0.93	6.00	<0.001	0.62, 1.23
	4.22	1.49	5.25	<0.001	0.93, 2.05
GLS					
	−3.71	−0.08	−0.96	0.338	−0.23, 0.08
	−0.71	−0.23	−4.18	<0.001	−0.33, 0.12
	3.28	−0.43	−4.76	<0.001	−0.61, −0.25

PPD: post-partum depression; UI: Unstable-Irregular; ET: Easy Temperament; LCC: lack of confidence in care; GLS: general level of safety.

## Data Availability

The data presented in this study are available on request from the corresponding author. The data are not publicly available due to privacy restrictions.

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
