# Peer review of "Is the Association between Postpartum Depression and Early Maternal–Infant Relationships Contextually Determined by Avoidant Coping in the Mother?"

_ijerph, 2021, doi:10.3390/ijerph18020562_

Round 1

Reviewer 1 Report

The theme of the manuscript is very relevant for the scientific community. However, the manuscript needs some modifications/corrections in order to be published.
Find below some suggestions:

Title: I suggest modifying the present title. In my opinion, it reflects the consequence of the study and not only the objective.

Summary: I suggest a complete review. It is not clear how the research will be carried out, especially about the method.

Introduction: the introduction reports depression in the perinatal period, so would the mother have it before or during pregnancy? I believe that this should be better explained, such as: What would be the factors that could influence avoidance coping in relation to postpartum depression?

Method: Describe, in detail, the participants (example: how were they recruited? What is the origin of them?). Which would be the inclusion and exclusion criteria? This article reports mothers with PPD, but the first assessment would be in the pregnancy. This measure should be better explained.
Describe the variables analyzed (for a better understanding of the results);

Results: The analysis performed in the first trimester of pregnancy would not be a measure of postpartum depression. This data should be reviewed;
I suggest that the Crombach value be placed in the results and more described (what the values mean). Was this analysis done in the global instrument or by parts that compose it?

Discussion: There are two measures of depression, one of which is not PPD. The first part of the result should be discussed in light of avoidance coping.

Conclusion: a brief conclusion of the research is necessary before the final considerations.

Reviewer 2 Report

Thank you to the editor and the authors for the opportunity to review this manuscript. The aim of the present study was to explore the moderating role of avoidant coping during early pregnancy in the relationship between PPD and maternal-baby relations. The strength of this study is that avoidant coping strategy was measured during pregnancy, which does not overlap with postpartum depression. Unfortunately, during pregnancy, the severity of depression was not checked at all. The authors assure that the women were mentally healthy, but do not state how they verified it. The results of the study are important from both a scientific and practical perspective. The article may be accepted for publication after important corrections have been made.

Major comments:

Line 1. The title of the article is misleading. It assumes an answer to the question of how we can improve the mother-child relationship in the case of depression. However, nowhere will the reader find an answer to this question.

Line 40 - Why authors cite articles based on the very specific populations (like Ghanian or Etiopian) when writing about PD in general?

Line 48 - Although the authors cite the work by Grace et.al (2003), they do not include their critical analysis in the paragraph. Grace analysis revealed that PPD has small effect on child development.

Line 96 - How was the mental health of the respondents verified? Since MD significantly influences the studied relationships (more than PDD, see Grace et. al (2003)), it is important how the mental health of the participants was determined.

Line 150-152 - please correct the reference referring to the fact that a blind review is not required.

Line 162 - Were there any differences in the studied variables between women who completed the EPDS and those who did not? There is a high attrition rate here, which means it is imperative to check if the study sample is not different than the whole group invited to the study.

Line 92 - The authors wrote that the research group consists of 116 women. Meanwhile, on line 162, they wrote that 111 women had completed their EPDS score. What is the reason for such a discrepancy?

Line 247 - It's not new for the literature that associations between PPD and maternal-infant relationships are not linear.

Line 327 - Conclusions must be completely changed. Although I personally share them, they do not result from the study. Authors cannot make assumptions or guesses in this part of the articles.

Reviewer 3 Report

My review on the paper “How can we improve early maternal-infant relationships in mothers with depressive symptoms? The role of avoidant coping” is the shortest I ever wrote, because there is really nothing important that is missing. The writing is fine, the explanations are well understandable, the methods are adequate. The only advice I have is that it could be interesting to include some thoughts what pattern of results one would have to expect in a more representative sample (including more mothers with higher PPD-scores). Apart from that I think this research is well conducted and the topic is interesting and timely.

Round 2

Reviewer 1 Report

All modifications suggested were made by the authors.

I congratulate the authors for the study and the manuscript.